# Numerical Simulation and Experimental Study on Residual Stress in the Curved Surface Forming of 12CrNi2 Alloy Steel by Laser Melting Deposition

**DOI:** 10.3390/ma13194316

**Published:** 2020-09-28

**Authors:** Zhaoxing Cui, Xiaodong Hu, Shiyun Dong, Shixing Yan, Xuan Zhao

**Affiliations:** 1School of Mechanical and Electronic Engineering, Shandong University of Science and Technology, Qingdao 266590, China; 15275266478@163.com (Z.C.); huxdd@163.com (X.H.); 2National Key Laboratory for Remanufacturing, Army Academy of Armored Forces, Beijing 100072, China; ysxneu@163.com; 3State Key Laboratory of Advanced Welding and Joining, Harbin Institute of Technology, Harbin 150001, China; zhaoxedu@163.com

**Keywords:** laser melting deposition, 12CrNi2 alloy steel, material parameters, thermal stress, residual stress

## Abstract

The performance and service life of the nuclear emergency diesel engine shaft made of 12CrNi2 alloy steel is very important for the safety of nuclear power. Laser melting deposition (LMD) is a challenging camshaft-forming technology due to its high precision, rapid prototyping, and excellent parts performance. However, LMD is an unsteady process under the local action of laser, especially for curved surface forming, which is more likely to generate large residual stress on components, resulting in cracks and other defects. At present, the stress research on LMD curved surface forming is relatively insufficient. In the present paper, material parameter testing, high-temperature mechanical properties analysis, single-track sample preparation, and heat source checks are conducted. At the same time, the ABAQUS software and the DFLUX heat source subroutine are used to compile the curved double-ellipsoidal moving heat source, and the effects of the temperature-dependent thermophysical parameters and phase change latent heat on the temperature field are considered. A three-dimensional finite element model is established to analyze the thermal stress evolution and residual stress distribution of multi-track multi-layer on a curved surface by LMD, and the effect of the scanning method and interlayer cooling time on the residual stress of the formed components is studied. The results show that with the increase in temperature, the strength of the material reduces, and the fracture morphology of the material gradually transitions from ductile fracture to creep fracture. The material parameters provide a guarantee for the simulation, and the errors of the width and depth of the melt pool are 4% and 9.6%, respectively. The simulation and experiment fit well. After cooling, the maximum equivalent stress is 686 MPa, which appears at the junction of the substrate and the deposited layer. The larger residual stress is mainly concentrated in the lower part of the deposited layer, where the maximum circumferential stress and axial stress are the tensile stress. Compared with the axial parallel lap scanning method, the arc copying lap scanning method has a relatively smaller maximum thermal stress and residual stress after cooling. The residual stress in the deposited layer is increased to some extent with the increase in the interlayer cooling time.

## 1. Introduction

12CrNi2 is a widely used low-carbon alloy carburized steel which is widely used in aerospace, electric power, petrochemical, marine, machinery, electronics, environmental protection, and other industries. The addition of nickel and chromium alloy elements can improve the hardenability of the steel and the strength and toughness of the carburized layer [1,2,3,4]. It is the preparation material for the camshaft of nuclear power emergency diesel engines [5,6]. As an important energy source, nuclear power plays an important role in human development and sustainable development [7]. A camshaft as an important component of a nuclear power emergency diesel generator; its performance and service life play a vital role in nuclear power safety [8,9]. At present, the camshaft is basically formed by traditional forging, casting, or a combination of parts, but due to the complex shape of the camshaft and the difficulty of machining, the traditional technology cannot meet the needs of the development of the camshaft manufacturing industry [10]. Laser melting deposition (LMD) technology is a new type of laser additive manufacturing (LAM) technology which has the advantages of high precision, a fast forming speed, excellent parts performance, and small machining margin, so it has become a challenging technology for camshaft manufacturing [11,12,13,14,15].

During the LMD process, based on the basic principle of rapid prototype manufacturing, the metal powder sent synchronously is melted to liquid metal by the heat source of a high energy laser according to the set processing path. Then, the liquid metal is quickly solidified, and the metal parts are directly formed through layer by layer deposition. The application prospect is very broad. However, LMD is an unsteady, extremely cold, and hot transient process. In the process of manufacturing accumulation, a local heat input will inevitably lead to a non-uniform temperature field. The local thermal effect is directly manifested as the solidification of the melt pool, and the residual stress is easily formed during the subsequent cooling process [16,17,18]. As a kind of internal stress, residual stress can directly affect the static load strength, fatigue strength, stress corrosion resistance, and dimensional stability of the formed parts. Therefore, the effective prediction of temperature change in the deposition process and the residual stress after cooling has important guiding significance for the performance control of components.

Numerical simulation is a convenient, fast, efficient, and economical method to predict the temperature and stress field of LMD, and it has been widely used [19,20]. Nazemi N et al. [21] established a finite element model of temperature history, microhardness value, and induced residual stress during the laser cladding of low/medium carbon steel plate P420 stainless steel powder, and studied the development of the residual stress of 10 single-track deposited specimens with different process parameters. Chew et al. [22] established a three-dimensional finite element model of the AISI 4340 steel laser cladding process, and simulated the residual stress distribution of single and multiple layers by laser cladding. Gong Cheng et al. [23] established a single-layer cladding model using the ANSYS software, simulated the temperature and stress field of 316L stainless steel during the cladding process, and studied the axial and horizontal residual stress distribution of the laser cladding layer. Alimardani et al. [24] proposed a three-dimensional transient finite element analysis method to simulate the thin-walled forming process of 304L stainless steel, and the transient temperature distribution of the melt pool and the real-time evolution of the stress field were obtained. Dai Deping et al. [25] used the Abaqus software to develop a nonlinear finite element method to simulate the temperature and stress field in the process of laser cladding. Using the developed calculation method, the temperature and stress field of the single-track single-layer cladding, single-track double-layer cladding, and single-track ten-layer cladding of Inconel718 Ni-base alloy were numerically simulated. Ding et al. [9] established a multi-layer multi-track finite element model using the ABAQUS software, and studied the effect of different substrate preheating temperatures on the residual stress of a 12CrNi2 multi-layer multi-track specimen. Kang et al. [26] established a single-layer, double-track finite element model of 24CrNiMo to simulate the temperature field and stress field distribution in the forming process of LMD. Kiran et al. [27] established a single-track, multi-track finite element model of 316L to simulate the temperature field and stress field distribution.

Gharbi et al. [28] studied the DMD process using a Yb-YAG disk laser and a widely used titanium alloy (Ti–6Al–4V) to understand the influence of the main process parameters on the surface finish quality. Qu et al. [29] fabricated a Ti-47Al-2.5V-1Cr intermetallic alloy by the laser melting deposition (LMD) manufacturing process and studied the microstructure by OM, SEM, TEM, and XRD. Cottam et al. [30] investigated the effect of the deposition rate on the residual stresses formed during the laser cladding of Ti-6Al-4V powder onto a Ti-6Al-4V substrate. The above studies are mainly aimed at titanium alloy [31], nickel-based alloy [32,33], stainless steel, and other materials [34], but there are relatively few studies on 12CrNi2 high-performance alloy steel, which plays an important role in the national economy and national defense. At the same time, the research on the temperature and stress field of LMD is mainly based on single-track, thin wall, and plane models. However, the camshaft forming process includes cam, boss forming, and many other curved surface forming processes, and the research on cam curved surface forming is still insufficient. The stress distribution of LMD curved surface forming is mainly divided into axial stress, circumferential stress, and radial stress, which are different from the transverse stress and longitudinal stress of plane in both the distribution and magnitude. How to realize the simulation of the temperature field and stress field of LMD curved surface forming to predict the stress distribution; how to understand the change in the mechanical properties of materials under high temperatures and the change in the thermal stress of the LMD curved surface forming process; and how to choose the appropriate forming strategy are the difficulties. In this paper, numerical simulation was combined with the experimental method. Based on the thermoelastic plastic method, the ABAQUS finite element software was used to compile the curved surface moving double-ellipsoid heat source program. Considering the influence of the thermophysical parameters changing with temperature and the latent heat of phase transition on the temperature field, the finite element analysis model of the curved surface multi-layer multi-track LMD process was established. The influence of the thermal stress evolution, residual stress distribution, scanning mode, and interlayer cooling time on the residual stress of the curved forming component was analyzed, which provided a reference for stress–strain regulation and the actual LMD curved surface forming and camshaft manufacturing.

## 2. Finite Element Model

### 2.1. Geometric Model

A schematic diagram of the curved surface forming by LMD is shown in Figure 1a. The round rod or round tube is chosen as the substrate constrained by a three-jaw chuck at one end and a thimble at the other. When the size of the substrate is long, support should be added in the middle of the substrate. In the forming process, the rotation of the substrate is driven by the three-jaw card; at the same time, the laser head moves in coordination under the control of a robot. Through the high-energy laser heat source, the metal powder sent synchronously to the robot is melted, quickly solidified, deposited layer by layer, and finally the metal component is formed. The forming strategy is shown in Figure 1b. At the beginning, the arc copying lap and S-scan mode were adopted.

Figure 2 shows the multi-layer and multi-track finite element model of the curved surface formed by LMD. The size of the substrate is 80mm in outer diameter, 10 mm in thickness, and 120 mm in length. The width of the deposited layer is 8mm and the thickness is 1.2 mm, which is composed of two layers and six tracks. The arc length of the deposited layer is 21 mm. The DC3D8 element for heat transfer is used in the temperature field. The sweep technique is used to divide the mesh. The deposition layer is divided into hexahedron mesh, and the mesh size is 0.5 mm × 0.5 mm × 0.3 mm. The mesh length of the substrate near the deposited layer is 1 mm, and it increases gradually to a maximum of 20 mm far away from the deposited layer. The mesh size becomes larger gradually in the thickness direction far away from the deposited layer. The minimum mesh is 0.2 mm and the maximum mesh is 2 mm.

In order to study the distribution law of temperature and stress, the data extraction path is drawn up as shown in Figure 3. Figure 3 shows the plane diagram after the curved surface is expanded, in which Node1 is the center point of the second track on the second deposited layer, path 1 is on the upper surface of the deposited layer, and path 2 is on the upper surface of the substrate. Both paths are in the same plane as Node1 and perpendicular to the scanning direction.

### 2.2. Finite Element Calculation Settings

The initial calculation model is based on the process parameters of laser power 2000 W, scanning speed 5 mm/s, and powder feeding rate 11.6 g/min. Aiming at the finite element calculation setting of the temperature field, firstly, according to the material parameters obtained by the test, the relevant material properties are set for the model, and the time step is mainly divided into three parts—namely, the element life and death step, the heat source loading step, and the cooling step. The main type is heat transfer. At the same time, heat transfer by convection and radiation is achieved by setting the convection and radiation boundary conditions, and the undeposited area is temporarily suppressed using the model change function. Secondly, we use the self-defined heat source subroutine to load and set the initial ambient temperature at 25 °C. Finally, we create a job to solve [35,36,37,38].

The finite element model of the stress field is the same as the temperature field model, but it is necessary to change the element type to 3D stress, and to set the prestress field as the odb file of the temperature field calculated before. Here, the right-end face of the model is restricted by five degrees of freedom, except the axial rotation, and the left-end face is imposed with axial and radial constraints.

### 2.3. Load and Boundary Conditions

#### 2.3.1. Heat Source Model and Latent Heat

As a kind of bulk heat source, a double-ellipsoidal heat source has been widely used in the calculation of temperature field, such as for welding and laser cladding, as it takes into account the inconsistent energy distribution caused by the movement of the heat source and the influence of the heat source on the depth direction. The laser energy is distributed in a certain volume and is applied to the nodes of the material model in the form of heat flux density. The double-ellipsoid heat source model is shown in Figure 4.

The heat flux inside the ellipsoid along the front half axis of the model is as follows:(1)$$q\left(x,y,z\right)=\frac{6\sqrt{3}f_{f}Q}{\pi \sqrt{\pi }abc_{f}}\exp\left\{-3\frac{x^{2}}{c_{f}{}^{2}}-3\frac{y^{2}}{a^{2}}-3\frac{z^{2}}{b^{2}}\right\}.$$

The heat flux inside the ellipsoid along the rear axis of the model is as follows:(2)$$q\left(x,y,z\right)=\frac{6\sqrt{3}f_{r}Q}{\pi \sqrt{\pi }abc_{b}}e\mathrm{xp}\left\{-3\frac{x^{2}}{c_{b}{}^{2}}-3\frac{y^{2}}{a^{2}}-3\frac{z^{2}}{b^{2}}\right\}.$$
where *Q* is effective thermal power; *a*, *b*, *c_f_*, and *c_b_* are heat source shape parameters; *f_f_* and *f_r_* are the proportional coefficients of the energy distribution of the anterior and posterior ellipsoid; *f_f_* + *f_r_* = 2. For the laser heat source, *c_f_* = *c_b_*= *a*, so *f_f_* = *f_r_*.
(3)Q=P⋅η,
where *P* is the laser power; η is the absorptivity of the material to the laser energy.

Therefore, the double-ellipsoidal heat source model can be expressed as:(4)$$q\left(x,y,z\right)=\frac{6\sqrt{3}P\eta }{\pi \sqrt{\pi }abc}e\mathrm{xp}\left\{-3\frac{x^{2}}{c^{2}}-3\frac{y^{2}}{a^{2}}-3\frac{z^{2}}{b^{2}}\right\}.$$

The above formula is based on the Cartesian coordinate system, which is very suitable for the plane state of the substrate surface, but for curved surface forming the size of the heat source is different and the temperature field is very unstable due to the change in spatial coordinates (x, y, z). Here, the double-ellipsoidal heat source is improved, taking the cylindrical coordinate system as the benchmark, considering the change with the spatial position, and introducing the *θ* angle. The formula is as follows:(5)$$q\left(x,y,z\right)=\frac{6\sqrt{3}P\eta }{\pi \sqrt{\pi }abc}e\mathrm{xp}\left\{-3\frac{{\left(x_{0}-R\cos\theta \right)}^{2}}{c^{2}}-3\frac{y^{2}}{a^{2}}-3\frac{{\left(z_{0}-R\sin\theta \right)}^{2}}{b^{2}}\right\},$$
(6)$$\theta =\frac{L}{R}-\theta_{0}=\frac{tv}{R}-\theta_{0}.$$
where *L* is the arc length of the heat source moving, and the size is the product of the scanning speed and the scanning time; *t* is the scanning time; *v* is the scanning speed; *R* is the curvature radius of the surface (when the surface is a cylindrical surface, *R* is the cylindrical radius); and $\theta_{0}$ is the angle between the line connecting the initial position to the center point of the surface and the positive direction of the X axis.

In the LMD process, there is a solid phase to liquid phase to solid phase transition, during which the material will continue to absorb or release a large amount of heat called phase transition latent heat. Therefore, the latent heat of the phase transition must be considered when establishing the finite element model [39]. The method of dealing with latent heat in ABAQUS (2017, SIMULIA, Johnston, RI, USA) is the enthalpy method—that is, the method of enthalpy varying with temperature is used to define latent heat, and its expression is as follows:(7)ΔH=∫ρc(T)dT,
where ΔH is enthalpy, ρ is density, and c is the specific heat capacity.

#### 2.3.2. Boundary Conditions

In the process of solving, it is necessary to give the initial temperature distribution and the boundary conditions of heat transfer in order to obtain the appropriate calculation results—that is, the uniqueness of the solution. There are three main types of boundary conditions [40,41,42]:

(1) The first kind of boundary condition:

Referring to the initial temperature or temperature function on the boundary of an object is the initial temperature distribution of the model before calculation. For the LMD process, it is mainly the temperature of the powder and substrate:(8)$$T|{}_{s}=T_{S}\;or\;T|{}_{s}=T_{S}(x,y,z,t),$$
where *S* is the boundary range of deposition forming, *T_S_* is the environment temperature, and $T_{S}(x,y,z,t)$ is the temperature function of the surface of the deposited specimen.

(2) The second kind of boundary condition:

Referring to the input of heat on the surface of the specimen and the known heat flux on the surface of the object is:(9)$$k_{x}\frac{\partial T}{\partial x}n_{x}+k_{y}\frac{\partial T}{\partial y}n_{y}+k_{z}\frac{\partial T}{\partial z}n_{z}=q_{s}(x,y,z,t),$$
where $n_{x}$, $n_{y}$, and $n_{z}$ are the cosine of the normal direction outside the boundary, and $q_{s}(x,y,z,t)$ is the heat flux density function.

(3) The third kind of boundary condition:

This refers to the heat exchange between the material model and the surrounding environment, which mainly includes convective heat transfer and thermal radiation. Among them, the convective heat transfer can be expressed as:(10)$$k_{x}\frac{\partial T}{\partial x}n_{x}+k_{y}\frac{\partial T}{\partial y}n_{y}+k_{z}\frac{\partial T}{\partial z}n_{z}=h(T_{e}-T_{s}),$$
where *h* is the convective heat transfer coefficient, *T_e_* is the fluid medium temperature, and *T_S_* is the environment temperature.

Thermal radiation can be expressed as:(11)$$k_{x}\frac{\partial T}{\partial x}n_{x}+k_{y}\frac{\partial T}{\partial y}n_{y}+k_{z}\frac{\partial T}{\partial z}n_{z}=\sigma^{\prime }\varepsilon (T_{e}{}^{4}-T_{s}{}^{4}),$$
where $\sigma^{\prime }$ is the Stephen Boltzmann constant, and the value is 5.67 × 10^−8^ W·m^−2^·K^−4^; ε is the radiation heat transfer coefficient.

In general, the boundary condition of the mixture of thermal convection and thermal radiation heat transfer is most often considered in the third kind of boundary condition, and its expression is as follows:(12)$$k_{x}\frac{\partial T}{\partial x}n_{x}+k_{y}\frac{\partial T}{\partial y}n_{y}+k_{z}\frac{\partial T}{\partial z}n_{z}=h(T_{e}-T_{s})+\sigma^{\prime }\varepsilon (T_{e}{}^{4}-T_{s}{}^{4}).$$

### 2.4. Stress Analysis

(1) Stress–strain relationship

In the process of LMD forming, the relationship between thermal elastoplastic stress and strain can be expressed as follows:(13){dσ}=[D]{dε}−{C}dT,
where {dσ} is the stress increment, {C} is the vector matrix related to the temperature, {dε} is the strain increment, [D] is the elastic-plastic or elastic matrix, and dT is the temperature increment.

If the material is in the elastic zone:(14)$$\left[D\right]={\left[D\right]}_{e},$$
(15)$$\{C\}={\{C\}}_{e}={\left[D\right]}_{e}\left(\{\alpha \}+\frac{\partial {\left[D\right]}_{e}^{-1}}{\partial T}\{\sigma \}\right),$$
where *T* is temperature and α is the linear expansion coefficient.

When the material reaches the plastic range:(16)$$f\left(\sigma \right)=f_{0}\left(\varepsilon_{p},T\right),$$
where $f_{0}$ is the yield stress function related to the temperature and plastic strain and f is the yield function.

At the same time, the mathematical expression of the plastic strain increment ${\{d\varepsilon \}}_{P}$ of the material is as follows:(17)$${\{d\varepsilon \}}_{p}=\lambda_{0}\{\frac{\partial f}{\partial \sigma }\},$$
where *λ_0_* is related to the hardening criterion and the material used for deposition.

(2) Solving process:

The relationship between the LMD numerical solution ${\left\{d\varepsilon \right\}}^{e}$ and ${\{d\delta \}}^{e}$ can be expressed as follows:(18)$${\left\{d\varepsilon \right\}}^{e}=\left[B\right]{\{d\delta \}}^{e}.$$

Finally, the stress increment {dσ} of the corresponding structural unit can be solved by the Formula (13).

## 3. Experimental Materials and Methods

### 3.1. Materials and Deposition Processes

The powder used in this experiment is 12CrNi2 alloy steel powder, and its chemical composition is shown in Table 1 [43].

After a lot of exploration and research from composition design to self-preparation, the powder was finally prepared by ultrasonic gas atomization equipment developed by the Institute of Metals, Chinese Academy of Sciences. 

A scanning electron microscope was used for observation, and the observation results are shown in Figure 5. It can be seen from the figure that the powder has a good sphericity, and most of the particles are regular spherical, which can form a better fluidity and is conducive to the powder’s absorption of laser energy.

The laser melting deposition experiment adopts the laser additive manufacturing system, which is composed of a laser system, powder feeding system, robot system, and control system. Among them, the laser system is mainly the IPG YLS-4000 (IPG, Burbach, Germany) fiber laser. The laser is output by a fiber with a 1mm core diameter, and the matching laser head is the PRECITEC YC52 (Precitec, Gaggenau, Germany) laser head. During the experiment, firstly, 80 sieve screens were used to screen the powder to filter impurities and large particles, and then the powder was put into an electric drying oven with a constant temperature. The heating temperature was set at 120 °C for 3 h to remove the moisture in the powder. Secondly, a muffle furnace is needed to preheat the substrate. Finally, the test can only be carried out on the premise of ensuring the normal operation of the protective gas and powder feeding system.

### 3.2. Testing of Material Parameters

#### 3.2.1. Testing of the Thermophysical Parameters of Materials

In order to ensure the accuracy of the thermophysical parameters of the material, the parameter data varying with temperature are obtained by experimental measurement and Jmatpro simulation. According to the experimental requirements, the sample size and test standards are shown in Table 2. Then, the scanning strategy of 90° interlayer rotation and the process parameters of the laser power of 2000 W, the scanning speed of 5 mm/s, and the powder feeding rate of 11.6 g/min are adopted to deposit the sample. The final sample size is obtained by wire cutting.

#### 3.2.2. Testing of Mechanical Properties at High Temperature

In order to test the high-temperature mechanical properties of 12CrNi2 alloy steel by laser melting deposition, the high-temperature tensile specimens were prepared in conjunction with the testing Center of University of Science and Technology Beijing. The high-temperature tensile tests were carried out according to the GB/T 228.2-2015 <<tensile Test of Metallic Materials-part 2: high temperature test method>> [45]. The size of the sample is shown in Figure 6.

### 3.3. Observation on the Morphology of the Molten Pool of the Sample

The process parameters of laser power 2000 W, scanning speed 5 mm/s, and powder feeding rate 11.6 g/min were selected to carry out a single-track deposition experiment. The sample was cut along the section by wire cutting, and the cut samples were inlaid by an automatic mosaic machine. Secondly, the embedded samples were soaked and washed in acetone, ground to 2000 mesh with sandpaper, polished, and then etched with 4% nitric acid alcohol solution for 10 s to make a metallographic sample. Finally, the prepared metallographic samples were placed under a three-dimensional profiler to observe the cross-section morphology of the melt pool.

## 4. Results and Discussion

### 4.1. Test Results of Material Parameters

#### 4.1.1. Thermophysical Parameters

The thermophysical parameters of the material are obtained as shown in Figure 7. It can be seen from the figure that with the increase in temperature, the specific heat capacity increases at first and then decreases, and the heat conduction coefficient decreases gradually, while the change in the thermal expansion coefficient fluctuates slightly, showing a trend of decreasing at first and then increasing as a whole.

#### 4.1.2. Mechanical Properties at High Temperatures

According to the GB/T 228.2-2015 standard [45], the parallel length is 40 mm, the extensometer standard distance length is 25 mm, the original standard distance length is 25 mm, and the parallel section diameter is 5 mm. In cooperation with the testing Center of University of Science and Technology of Beijing, the tensile properties of the samples at 200, 400, 600, and 800 °C are tested. According to the tensile data obtained, the stress–strain curves are drawn as shown in Figure 8.

Figure 8 shows that with the increase in temperature, the yield strength decreases gradually, while the tensile strength increases at first and then decreases, and the elongation after fracture increases. The increase in temperature reduces the strength and improves the plastic properties of the material.

The classical plasticity theory of metal materials is applied for the default plastic material characteristics of ABAQUS, and mises yield surface is used to define the isotropic yield. The plastic deformation behavior of metal materials can be briefly described, as shown in Figure 9. In the case of small strain, the material properties are basically linear elastic and the elastic modulus E is constant; when the stress exceeds the yield stress, the stiffness decreases significantly, and the strain of the material includes plastic strain and elastic strain. After unloading, the elastic strain disappears, but the plastic strain is irrecoverable; if loaded again, the yield stress of the material will increase, which is the so-called work hardening.

The data obtained in the uniaxial tensile experiment are usually expressed in terms of nominal strain and nominal stress. Plastic strain and real stress are required when defining plastic material parameters in ABAQUS. Therefore, they are converted according to relevant theories, and the elastic modulus and yield strength of the material are calculated as shown in Figure 10, which shows that the elastic modulus and yield limit of the material decrease gradually with the increase in temperature.

Figure 11 shows the surface morphology of the tensile specimens. An obvious necking phenomenon is observed in the specimens at 200, 400, and 600 °C, and the fracture surface of the sample at 800 °C is zigzag. With the increase in temperature, the surface oxidation of the sample is gradually intensified, and the length of the sample after fracture is gradually increased.

The fracture morphology was further observed by scanning electron microscope, as shown in Figure 12 and Figure 13. Both 200 and 400 °C are ductile fractures. Figure 12a,c shows that there are fiber zones, shear lip zones, and a small amount of pore defects on the macroscopic fracture surface of the samples at 200 and 400 °C. There is a large-sized hole in the center of the sample at 200 °C, which is the result of the exfoliation of unmelted powder or large-sized inclusions in the separation process of the fracture sample under tensile load. In Figure 12b,d, there are dimples. In addition, a large number of micropores and small dimples can be seen in Figure 12b, which is an obvious feature of a micropore accumulation fracture. The dimples of the samples at 200 °C are smaller and shallower than those at 400 °C.

Figure 13 shows the fracture morphology of the samples at 600 and 800 °C. The creep fracture occurs at 600 and 800 °C, which is due to the creep phenomenon during tension at high temperatures. Figure 13a,c shows that the macroscopic fracture surfaces of the samples at 600 and 800 °C are not smooth, in which the specimen at 600 °C has a necking phenomenon and the creep fracture has obvious plastic deformation at the same time, which is the result of the transgranular propagation of creep crack. Meanwhile, there is no necking phenomenon in the specimen at 800 °C, but there is obvious plastic deformation at the time of creep fracture, which is caused by the propagation of creep cracks spreading along the grain boundary.

### 4.2. Experimental Results and Model Verification

#### 4.2.1. Experimental Results

Figure 14 shows the schematic diagram of single-channel deposition. A “binding zone” is formed at the interface between the deposited layer and the substrate, and the formation of this part makes a good metallurgical bond between the substrate and the deposited layer. The surface of the molten pool is arc-shaped, and the substrate is partially melted to form a certain depth of penetration. At the same time, due to the concentration of laser energy, the heat-affected zone after the action of the heat source is very small, and the white edge line in the picture is the boundary of the heat-affected zone. The width of the heat-affected zone basically tends to a 0.5 mm range.

#### 4.2.2. The Heat Source Check

In order to obtain more reasonable heat source parameters and ensure the accuracy of the finite element numerical simulation of the temperature field, the finite element model of the temperature field of the curved single layer and single track is established. As shown in Figure 15, the substrate selects a round rod with the same size as the central shaft of the camshaft, with diameter of 80 mm and length of 120 mm. The size of the deposited layer is consistent with the size of the single-track section prepared under the parameters with the laser power of 2000 W, the scanning speed of 5 mm/s, and the powder feeding rate of 11.6 g/min. A quarter of the circumference of the central shaft is chosen as the arc length of the deposited layer. A proper simplification has been made here. Consider that the single-track deposition on the curved surface can be regarded as thick-plate deposition, and the size and morphology of the molten pool obtained are almost the same as those of the plate deposition. Therefore, using the experiment of plate deposition instead of the curved deposition here can not only save the experimental cost, but also obtain accurate verification results. The DC3D8 thermal analysis unit is selected as the element type, and the process parameters of a laser power of 2000 W, scanning speed of 5 mm/s, and powder feeding rate of 11.6 g/min are used for the simulation. The simulated temperature-field molten pool cloud image is compared with the molten pool morphology of the previous actual process experiment to verify the accuracy of the simulation. The simulated cloud image of the melt pool is compared with the cross-section morphology of the melt pool obtained in the experiment above to verify the accuracy of the simulation.

Figure 16 shows the comparison of the melt pool obtained through simulation and experiment. The dotted line in the figure represents the melting point line of alloy steel, whose value is 1470 °C. When the temperature is higher than this temperature, the substrate and powder melt to form a melt pool. The width of the melt pool obtained by the experiment is 3.46 mm and the depth is 1.04 mm. The width of the simulated melt pool is 3.32 mm and the depth of the melt pool is 0.94 mm, with errors of 4% and 9.6%, respectively. The experiment and simulation fit well, and the data are shown in Table 3. The causes of the errors may be as follows: (1) the high-temperature thermophysical parameters are difficult to measure directly due to the limitation of conditions. At the same time, the data obtained by the Jmatpro software simulation and experimental test have errors. (2) Error exists between the heat transfer conditions in the numerical simulation and the actual experimental environment; (3) the effects of the material gasification and melt pool flow on the heat transfer are ignored in the simulation.

### 4.3. Thermal Stress Evolution

Figure 17 shows a schematic diagram of the thermal stress induced by the temperature gradient. In the process of LMD, a laser beam rapidly heats the powder on the upper surface of the substrate at room temperature. The high-energy heat source will quickly melt the powder to form a melt pool, and with the heat source moving, the melt pool will rapidly cool and solidify. Because the laser beam heats the upper surface layer rapidly and the heat conduction is slow, the temperature gradient rises sharply. With the increase in temperature, the strength of the material decreases at the same time, and the expansion of the top heating layer is limited by the elastic compression of the lower material. When the yield strength of the material is reached, the top layer will be plastically compressed, inducing strain and producing compressive stress. In the absence of external mechanical constraints, reverse bending can occur in the direction away from the laser beam. In the subsequent cooling process, the upper layer of the plastic compression begins to contract and bend at a certain angle to one side in the direction of the laser beam, but due to the limitation of the base material at the bottom, the upper layer that tends to bend is constrained by the substrate at the bottom to produce tension, while the lower layer is subjected to pressure. Therefore, tensile stress can be formed in the deposited layer, and compressive stress is mainly formed in the lower part of the deposited layer.

Figure 18 shows the temperature field distribution when scanning to the center of the second track of the first layer. The peak temperature at this time is 2347 °C, and the front isotherm is denser than the back end in the scanning direction, which is due to the inconsistent energy of the melt pool caused by the movement of the laser. By setting the temperature display limit to the melting temperature of alloy steel at 1470 °C, the width and depth of the melt pool can be shown in the temperature field cloud picture. In the early stage, the simulation has been fitted with the experiment, and the fitting is good.

Figure 19 shows the cloud map of the thermal stress distribution during deposition. At the position of laser action, the material is in the melting state, and the equivalent stress is very small and approaches 0 MPa. When the laser moves forward far away from the region, the temperature there gradually decreases, and the melt pool gradually solidifies and shrinks. Meanwhile, the shrinkage will be prevented by the solidified metal nearby, which will exert a certain constraint effect and gradually form thermal stress. The magnitude of the thermal stress mainly depends on the temperature gradient. When the temperature gradient is too large, the thermal stress will exceed the yield limit of the material and produce plastic deformation. With the gradual decrease in temperature, the thermal stress will gradually become stable, and the residual stress after cooling is finally formed.

Figure 20 shows the thermal stress evolution curve of the deposition process of Node1. When the heat source moves to the selection point, the thermal stress approaches zero and then gradually increases to about 110 MPa with the departure of the heat source. Due to the remelting effect between adjacent tracks, when this position is melted by the adjacent heat source again, the thermal stress approaches 0 again, then increases gradually with the decrease in temperature and finally stabilizes at about 350 MPa after cooling.

### 4.4. Residual Stress Distribution

In the initial stage of LMD, the substrate is at a low temperature, and the upper powder is melted by the high-energy heat source rapidly, resulting in a very large temperature gradient in the bonding area. With the progress of laser deposition, the temperature of the substrate rises gradually and the heat accumulates, leading to a gradual decrease in the temperature gradient until the temperature gradient at the top is the smallest. The greater the temperature gradient is, the greater the residual stress is. Therefore, the residual stress is greatest in the region where the deposited layer and the substrate are combined.

Figure 21 is a cloud map of the residual stress distribution after cooling. The maximum equivalent stress is 686 MPa, which appears at the junction of substrate and deposition, where a large temperature gradient is easily formed. Due to the difference in the thermal expansion coefficient, different expansions and shrinkages of metal after cooling will be formed, resulting in greater stress and deformation. The circumferential residual stress is symmetrically distributed relative to the scanning direction (ROT plane), and the maximum stress is 719 MPa. At the same time, the axial residual stress is symmetrically distributed relative to the ROZ plane, and the maximum stress is 751 MPa. Furthermore, the axial residual stress is slightly larger than the circumferential residual stress. The reason is that axial deformation is restricted by axial constraints, which makes it difficult for materials to expand and shrink, and thus greater residual stress is generated.

Figure 22 shows the nephogram of residual stress in the direction of the ROZ section. The greater residual stress is mainly concentrated near the heat-affected zone, where the maximum circumferential stress and axial stress are tensile stress, while compressive stress is formed not far below. The overall residual stress of the deposited layer is relatively small.

Figure 23 shows the residual stress distribution curve of the extracted path 1 and path 2. Figure 23a shows that along path 1, the mises stress and circumferential stress on the deposited layer increase at first and then decrease. The maximum residual stress appears near the last path, and the residual stress on both sides is relatively small. The main reason is that this position is the last position of deposition, where a large heat accumulation and temperature gradient are formed. However, the stress in the former position of deposition will be partially released due to the remelting and post-thermal action of post-deposition. Furthermore, the circumferential stress shows tensile stress, and the maximum value is about 305 MPa, while the axial stress gradually changes from compressive stress to tensile stress, and the maximum stress is about 90 MPa.

Figure 23b shows the residual stress distribution curve of path 2 on the upper surface of the substrate. The change trend of the curve is similar to that of the residual stress curve in welding—that is, the circumferential residual stress in the deposited region shows a large tensile stress, with a maximum value of about 455 MPa. Extending from the deposited layer to both ends, the residual stress decreases gradually and finally tends to compressive stress, of which the maximum value is about 160 MPa. At the same time, the axial residual stress is mainly tensile stress, which is small in the deposition area and reaches a maximum value at about 630 MPa at the junction between the deposited layer and the substrate. With the distance away from the deposited layer, the stress decreases gradually.

### 4.5. The Influence of Scanning Mode on Stress

The camshaft formed by LMD is divided into two steps: the first step is to form the central shaft, and the second step is to deposit the cam and convex platform on the central shaft. For the curved surface deposition of the cam and convex platform, two forming methods can be considered: an arc copying lap and an axial parallel lap. As shown in Figure 24, the main difference between the two methods is that the single-track direction is not the same, so that the length of the single track is not the same. The arc length of this layer is selected as the single-track length of the method of the arc copying lap, and the lap direction is parallel to the motion friction direction of the cam, which is beneficial to improve the formability and service life of the cam. In the axial parallel lap-forming mode, the single-track length is the cam thickness and the arc length is short. The cam thickness is chosen as the single-track length of the axial parallel lap forming, and the arc length is shorter.

Figure 25 shows the residual stress distribution of path 1 on the surface of the deposited layer with different scanning modes. The mises stress, circumferential stress, and axial stress increase at first and then decrease along the positive direction of the Z axis. A large residual stress is formed with the axial parallel lap scan on the deposited layer because, compared with the arc copying lap scan, the axial parallel scan produces more deposition trajectories and the turning and pause times of the laser heat source, so that the heat source is unstable, which results in a larger cooling rate and temperature gradient, and finally a greater stress.

Figure 26 shows the thermal stress change curve of the first track central point of the first layer with different scanning modes. During the movement of the laser heat source, the thermal stress of this point is constantly changing and is reduced with each approach of the heat source, which is consistent with the mechanical properties at high temperatures. Meanwhile, as the heat source leaves, the thermal stress increases again. Compared with the axial parallel scan, the maximum thermal stress in the process and the residual stress of arc copying lap scan are relatively small.

### 4.6. Effect of Interlayer Cooling on Residual Stress

Figure 27 shows the residual stress distribution of path 1 on the surface of the deposited layer with the action of the interlaminar cooling time. The change trend of residual stress is about the same. The mises stress and circumferential stress increase at first and then decrease along the Z axis, while the axial stress decreases at first and then increases. With the increase in the interlaminar cooling time, the maximum stress on the deposited layer gradually increases, while the maximum mises stress increases from 335 to 531 MPa, and the maximum circumferential stress increases from 301 to 503 MPa. The residual stress on the upper surface of the deposited layer increases to a certain extent with the increase in the interlayer cooling time, which is mainly because, with the increase in the interlayer cooling time, the temperature of the initial layer decreases and the re-deposition forms a larger temperature gradient so that the stress increases. However, in the actual forming process, ensuring a certain interlaminar cooling is necessary. Continuous deposition produces heat accumulation, which increases the size of the melt pool and is not conducive to forming.

## 5. Conclusions

The main results are as follows: 

(1) The strength of the material decreases and the plastic property increases with the increase in temperature. In the forming process of the LMD curved surface, a large instantaneous thermal stress will be generated where the temperature gradient is large, and when the thermal stress is higher than the strength of material at this temperature the thermal cracks are easily generated. The necking phenomenon appeared in the samples at 200, 400, and 600 °C, and the fracture surface of the sample at 800 °C was sawtooth. With the increase in temperature, the surface oxidation of the sample is gradually intense, and the post-fracture length of the sample increases gradually. Ductile fractures occur at 200 and 400 °C, and creep fractures occur at 600 and 800 °C.

(2) The test material parameters provide the premise and guarantee for the simulation. The size of the melt pool obtained by the LMD numerical simulation of the curved single layer and single channel is compared with the melt pool size of the experimental test. The errors of the width and depth of melt pool are 4% and 9.6%, respectively, and the simulation fits well with the experiment.

(3) The thermal stress of the LMD curved surface forming is mainly divided into axial stress, radial stress, and circumferential stress, which are different from the stress in the plane state in distribution and magnitude. During the LMD process, the thermal stress varies with time. The maximum equivalent stress after cooling is 686 MPa, which appears at the junction of the substrate and deposition layer. The circumferential residual stress is symmetrically distributed relative to the scanning direction (ROT plane) and the maximum stress is 719 MPa, while the axial residual stress is symmetrical relative to the ROZ plane and the maximum stress is 751 MPa. The larger residual stress is mainly concentrated in the lower part of the sedimentary layer, where the maximum circumferential stress and axial stress are tensile stress, while the compressive stress is formed not far below. The performances and crack defects of the components are easily effected by tensile stress, a harmful stress, which should be reduced or eliminated as much as possible.

(4) Compared with the axial parallel lap scanning mode, the maximum thermal stress and the residual stress after cooling are relatively smaller in the arc copying lap scanning mode. Meanwhile, taking into account the advantages of being less time-consuming and having a lower path turning time, this is an ideal forming method. The increase in the interlayer cooling time can increase the residual stress on the upper surface of the deposited layer to some extent, which needs to be avoided. However, certain interlayer cooling is helpful to reduce heat accumulation and facilitate the forming of components. Therefore, a small interlayer cooling time is selected to obtain a good formability and produce a small residual stress, which is an optimal strategy.

## Figures and Tables

**Figure 1 materials-13-04316-f001:**
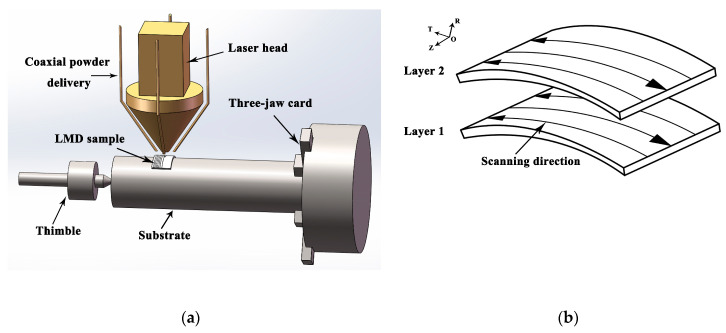
Schematics of (**a**) the LMD curved surface forming and (**b**) scanning strategy.

**Figure 2 materials-13-04316-f002:**
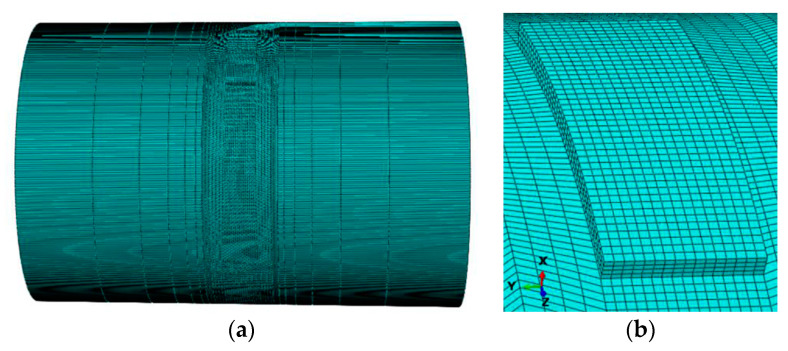
Mesh model: (**a**) overall model, (**b**) partial model.

**Figure 3 materials-13-04316-f003:**
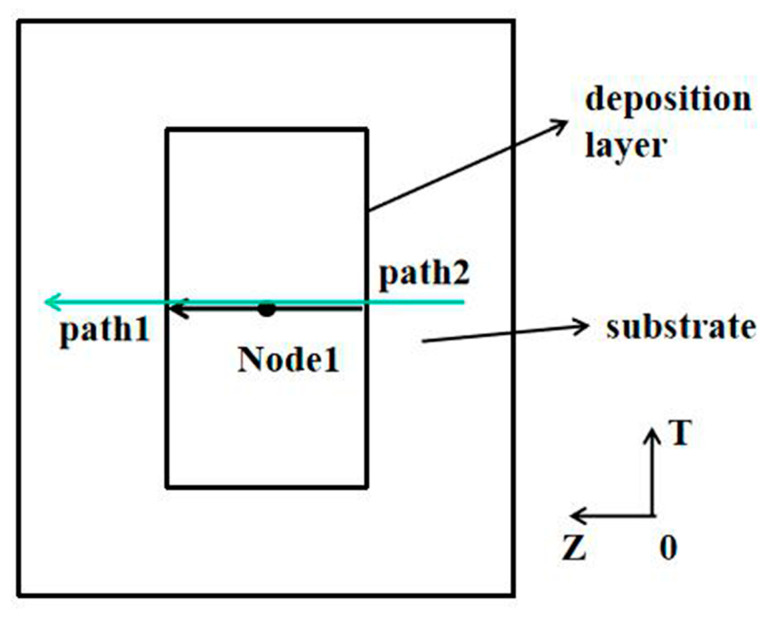
Schematic diagram of the data extraction location.

**Figure 4 materials-13-04316-f004:**
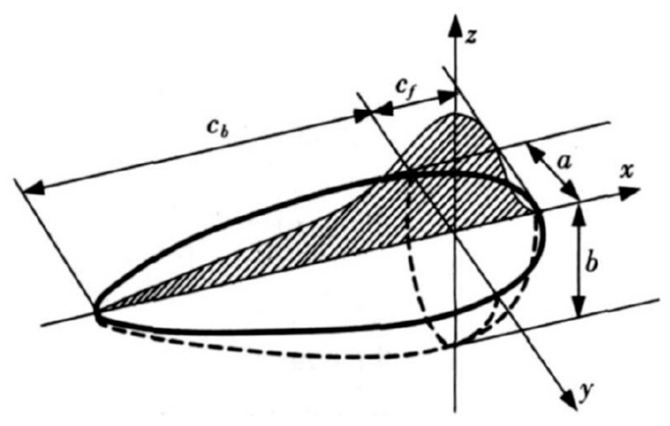
Double-ellipsoid heat source model.

**Figure 5 materials-13-04316-f005:**
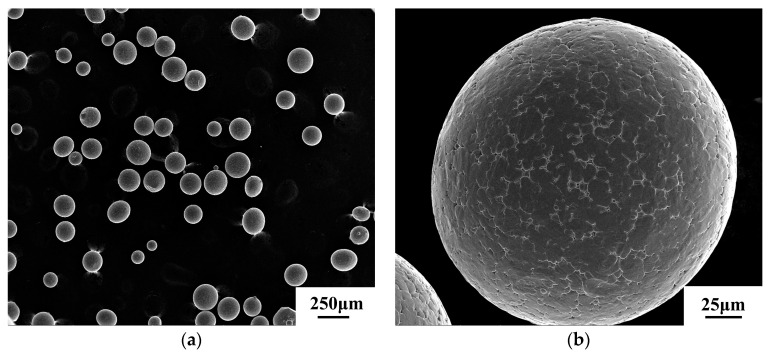
Micromorphology of the 12CrNi2 alloy steel powder: (**a**) multiple powder, (**b**) single powder.

**Figure 6 materials-13-04316-f006:**
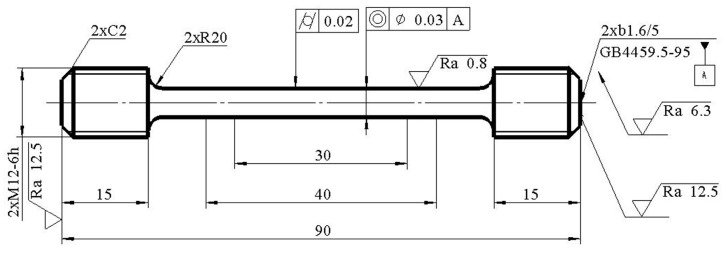
Dimensional drawing of the high-temperature tensile specimen (unit: mm).

**Figure 7 materials-13-04316-f007:**
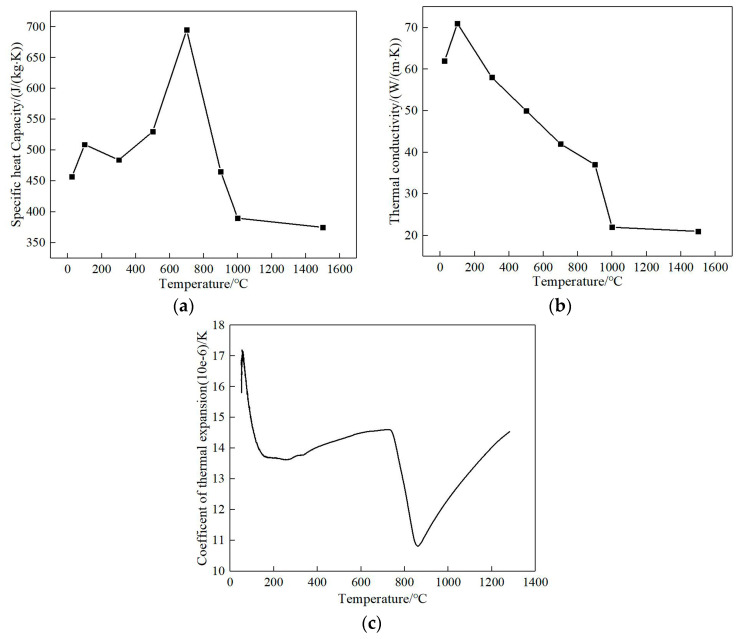
Physical parameters of 12CrNi2: (**a**) specific heat, (**b**) thermal conductivity, (**c**) coefficient of thermal expansion.

**Figure 8 materials-13-04316-f008:**
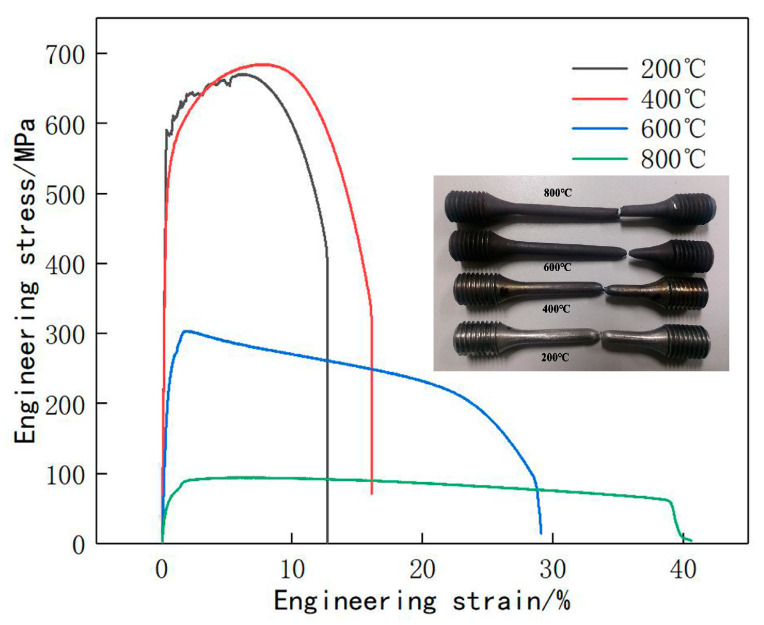
Stress–strain curve of 12CrNi2.

**Figure 9 materials-13-04316-f009:**
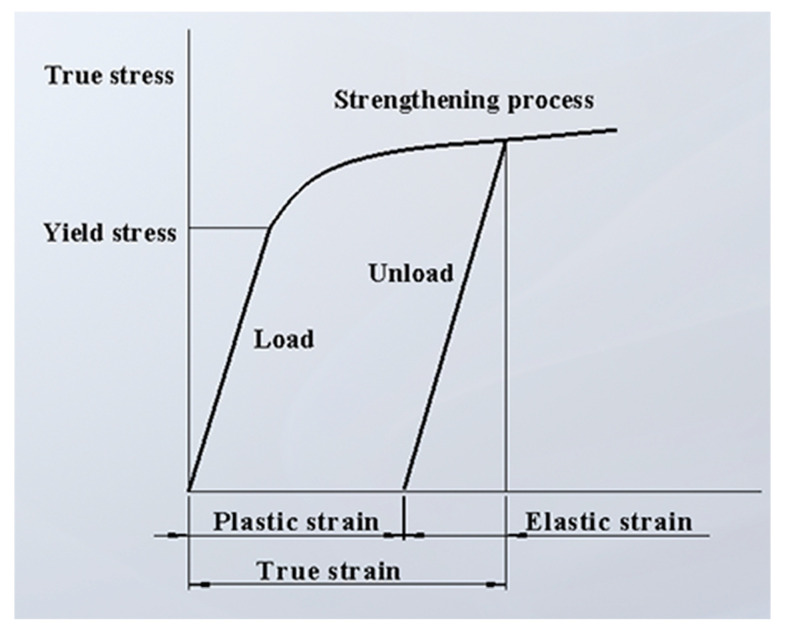
Real stress–strain curve.

**Figure 10 materials-13-04316-f010:**
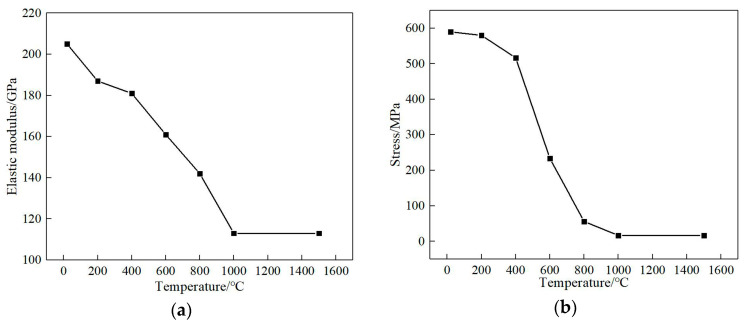
Physical parameters of 12CrNi2: (**a**) modulus of elasticity, (**b**) yield limit.

**Figure 11 materials-13-04316-f011:**
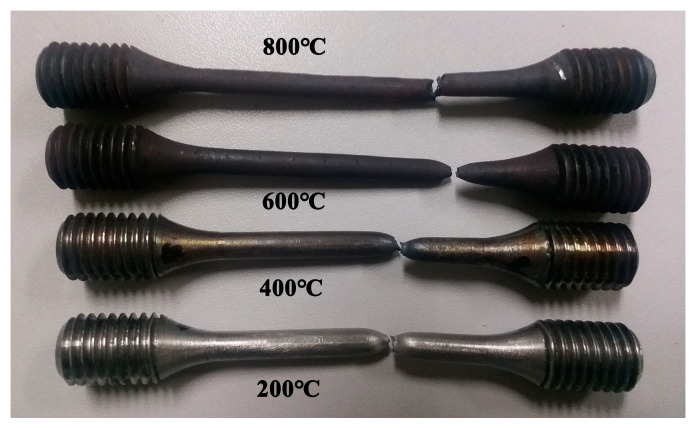
Surface morphology of the samples.

**Figure 12 materials-13-04316-f012:**
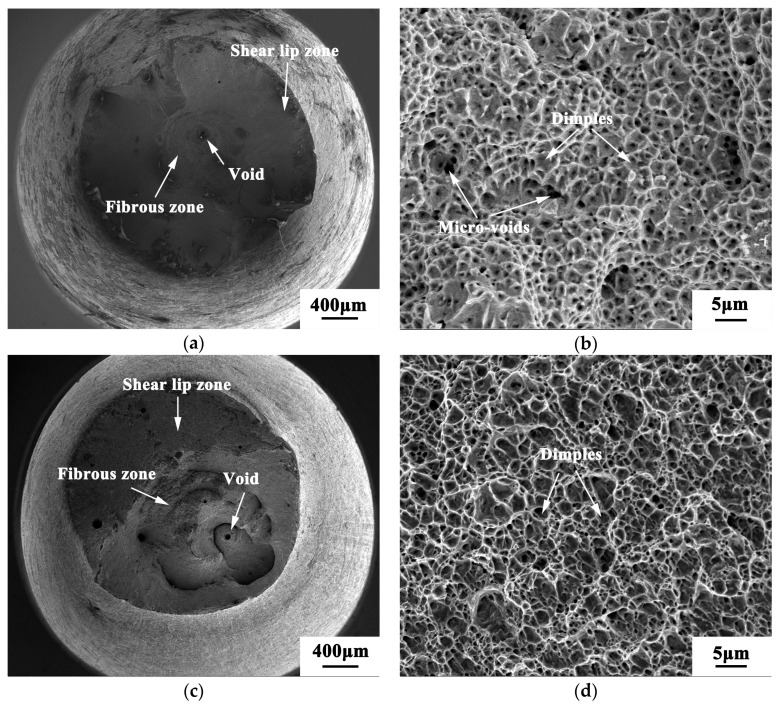
Fracture morphology at 200 and 400 °C: (**a**,**b**) 200 °C, (**c**,**d**) 400 °C.

**Figure 13 materials-13-04316-f013:**
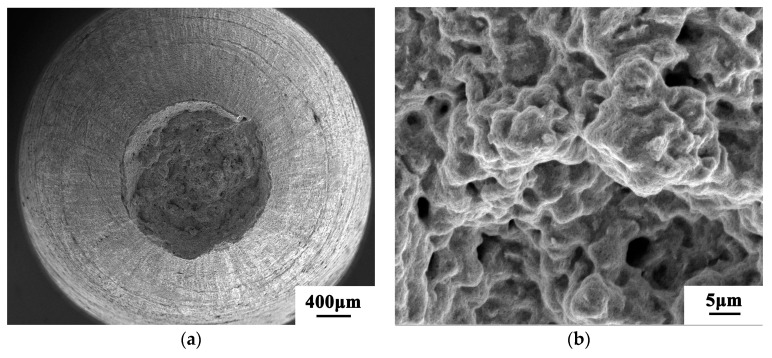
Fracture morphology at 600 and 800 °C: (**a**,**b**) 600 °C, (**c**,**d**) 800 °C.

**Figure 14 materials-13-04316-f014:**
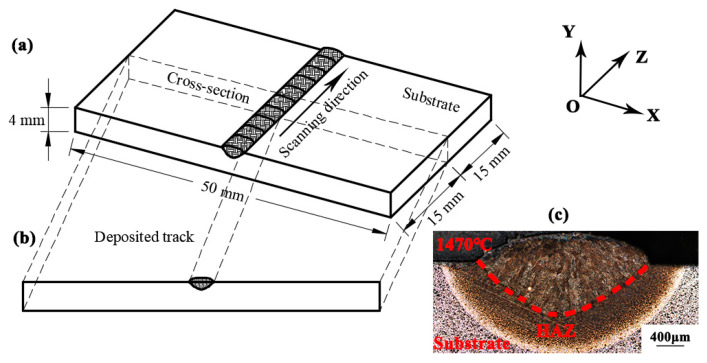
The schematic diagram of: (**a**) single-channel deposition, (**b**) the cross-section of the single-channel deposition, (**c**) the cross-section morphology of the single-channel deposition.

**Figure 15 materials-13-04316-f015:**
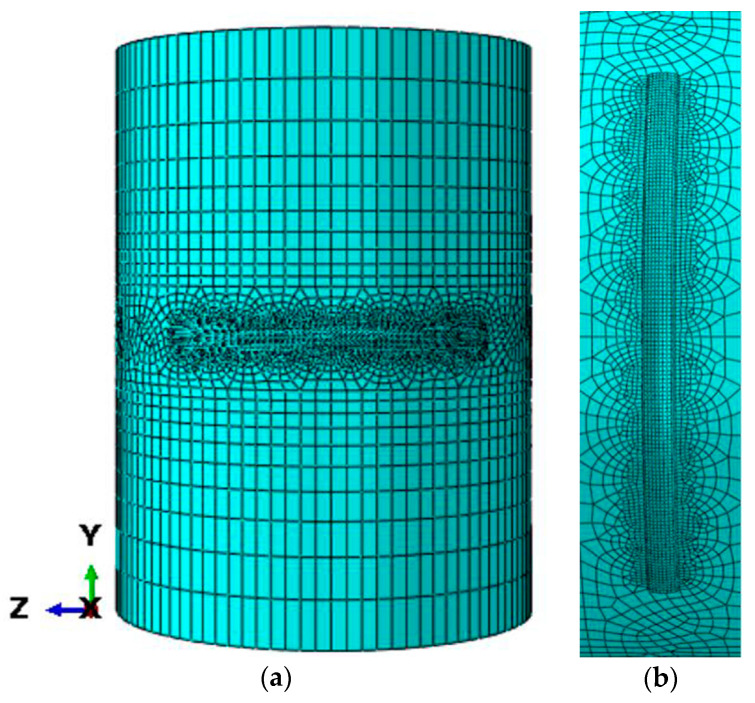
Single-track geometric model: (**a**) overall model, (**b**) partial model.

**Figure 16 materials-13-04316-f016:**
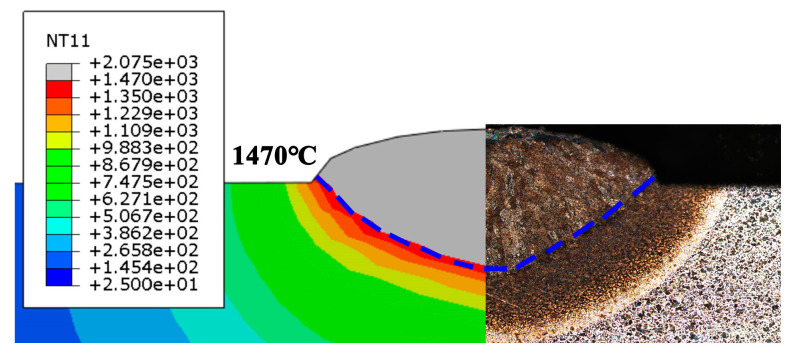
Comparison of the melt pool obtained by the simulation and experiment.

**Figure 17 materials-13-04316-f017:**
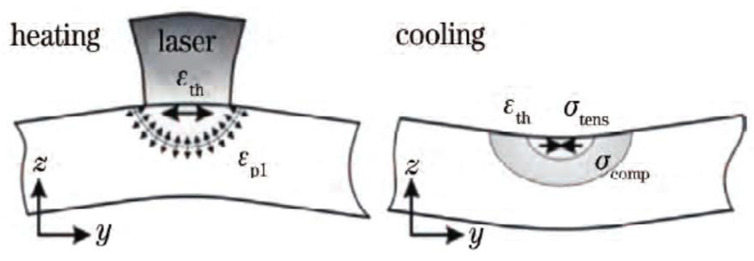
Thermal stress induced by the temperature gradient.

**Figure 18 materials-13-04316-f018:**
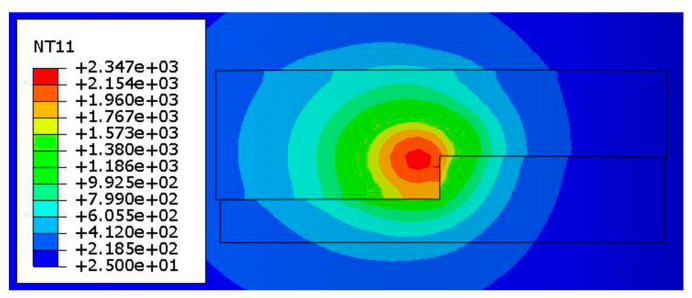
Temperature field distribution when scanning to the center of the second layer of the first layer.

**Figure 19 materials-13-04316-f019:**
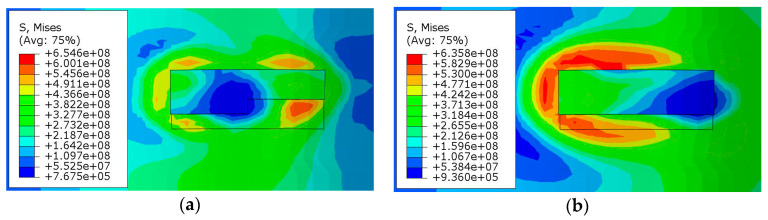
Thermal stress distribution during deposition: (**a**) scan to the center point of the first layer of track 2; (**b**) scan to the end of the first layer of track 2.

**Figure 20 materials-13-04316-f020:**
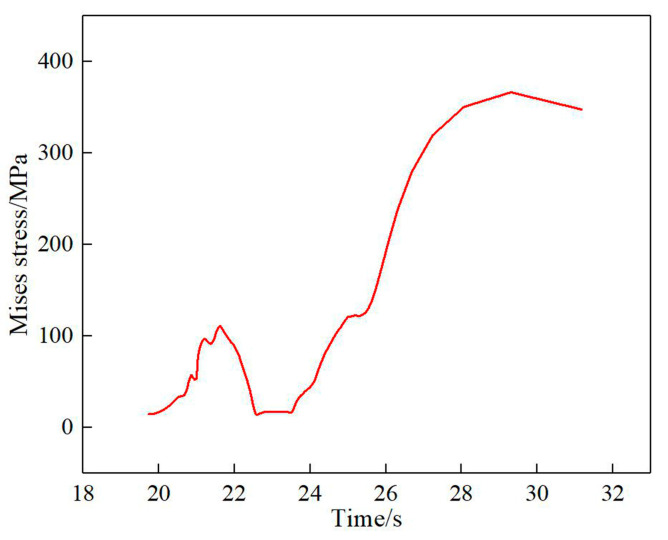
Thermal stress evolution during the deposition of Node1.

**Figure 21 materials-13-04316-f021:**
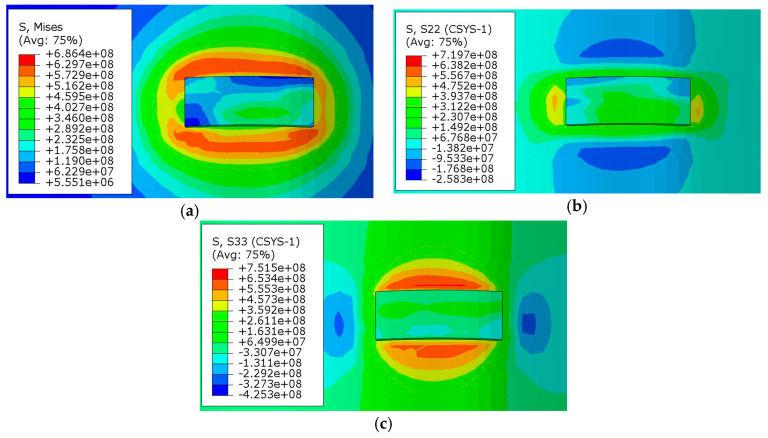
Residual stress distribution after cooling: (**a**) mises stress, (**b**) circumferential stress, (**c**) axial stress.

**Figure 22 materials-13-04316-f022:**
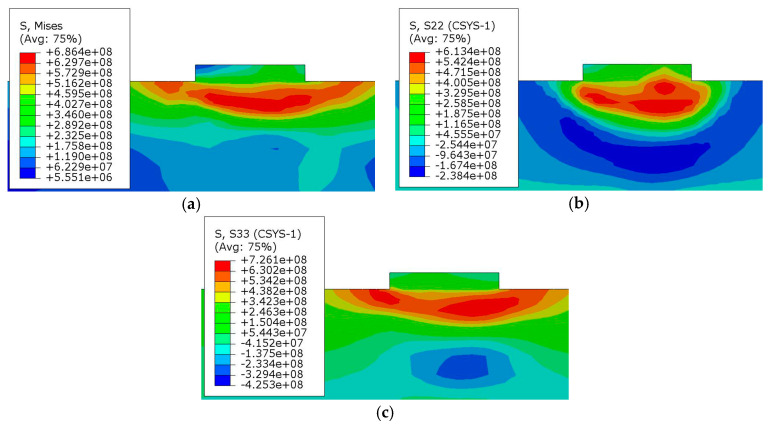
Residual stress distribution in the section direction: (**a**) mises stress, (**b**) circumferential stress, (**c**) axial stress.

**Figure 23 materials-13-04316-f023:**
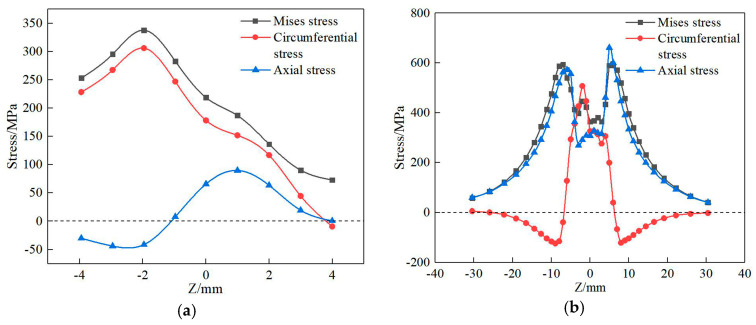
Residual stress distribution on a path perpendicular to the scanning direction: (**a**) path 1, (**b**) path 2.

**Figure 24 materials-13-04316-f024:**
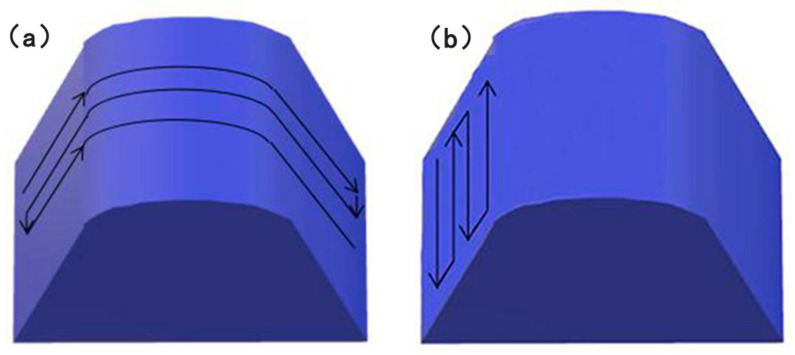
Top view of the forming method of cam: (**a**) arc copying lap, (**b**) axial parallel lap.

**Figure 25 materials-13-04316-f025:**
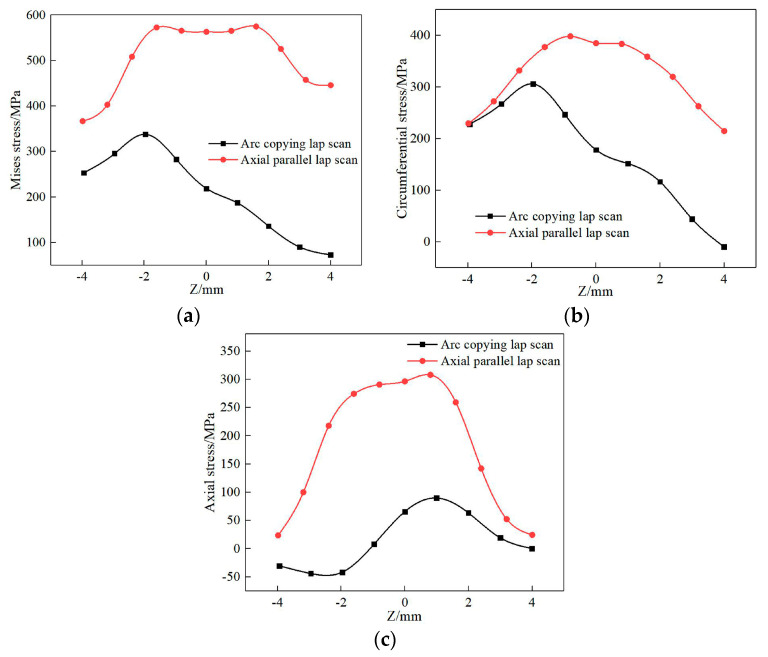
Stress distribution of the different scanning methods on path 1: (**a**) mises stress, (**b**) circumferential stress, (**c**) axial stress.

**Figure 26 materials-13-04316-f026:**
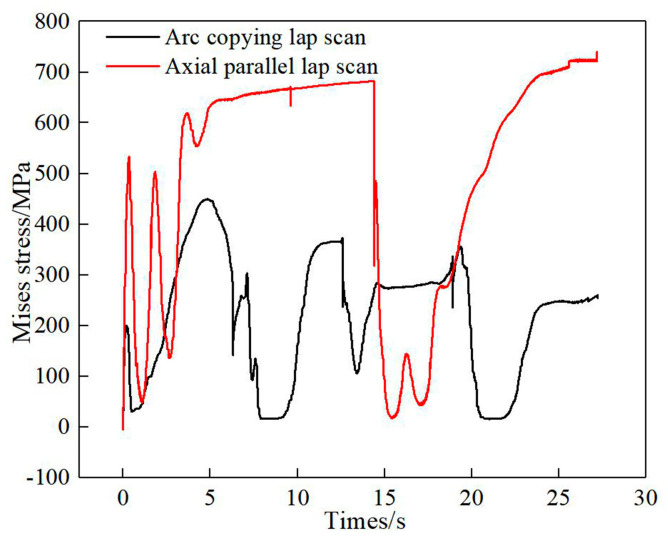
Thermal stress curves of the first track central point of the first layer with different scanning modes.

**Figure 27 materials-13-04316-f027:**
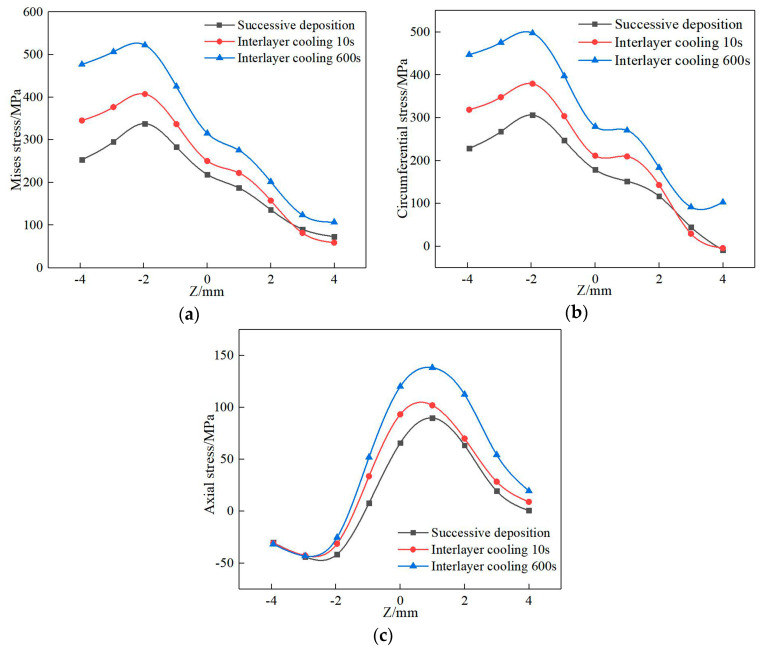
Stress distribution of the different interlayer cooling times of path 1: (**a**) mises stress, (**b**) circumferential stress, (**c**) axial stress.

**Table 1 materials-13-04316-t001:** Chemical composition of the 12CrNi2 powder (quality components, wt%).

Element	C	Fe	Si	Cr	Ni	Mn	O
Content	0.12	Bal	0.34	1	1.59	0.57	0.008

**Table 2 materials-13-04316-t002:** The sample size and test standards of the different tested parameters.

Parameter Name	The Sample Size	Test Standards or Equipment
The thermal conductivity and specific heat capacity	φ12.5 mm × 2.5 mm	GB/T22588-2008 [44]
Density	φ15 mm × 35 mm	UAE/PH-TDWN010 automatic true density analyzer
The thermal expansion coefficient	φ3 mm × 50 mm	Thermal expansion analyzer (Baehr, Pirmasens Germany)

**Table 3 materials-13-04316-t003:** Comparison of the size of the melt pool obtained by the simulation and experiment.

Name	Experimental Value	Simulation Value	Error
The width of melt pool/mm	3.46	3.32	4%
The depth of melt pool/mm	1.04	0.94	9.6%

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
