# Peer review of "Numerical Simulation and Experimental Study on Residual Stress in the Curved Surface Forming of 12CrNi2 Alloy Steel by Laser Melting Deposition"

_materials, 2020, doi:10.3390/ma13194316_

Round 1

Reviewer 1 Report

The work is original and relevant to scope of journal because it considered the comparative analysis of numerical simulation and experimental study on residual stress in curved surface forming of 12CrNi2 alloy steel by laser melting deposition. It is conducted with the help of new empirical methods - Finite Element Model. The research method are explained fully.

The results are properly discussed and compared to the previous studies. Data is included and properly analyzed.

In this paper, authors used 39 sources, containing both historical and fundamental works, as well as the latest scientific research on this topic. But the literature review can be improved.

For example, the papers of Yumashev discussed many point of this study:

1.Yumashev, A., Ślusarczyk, B., Kondrashev, S., Mikhaylov, A. (2020). Global Indicators of Sustainable Development: Evaluation of the Influence of the Human Development Index on Consumption and Quality of Energy. Energies, 13, 2768. https://doi.org/10.3390/en1311276

The authors have provided opportunities for further research and writing scientific articles based on the data obtained.

It is right conclusion that the strength of the material decreases and the plastic property increases with the increase of temperature. In the forming process of LMD curved surface, large instantaneous thermal stress

will be generated where the temperature gradient is large, and when the thermal stress is higher than the strength of material at this temperature, the thermal cracks are easy generated. The findings are in potential interest to material engineers.

The paper possesses a proper form of well-structured and readable technical language of the field and represents the expected knowledge of the journal`s readership. There are minor errors in English, this affects positively the general nature of the work.

The current study brings many new to the existing literature or field. For one, the author(s) seem to have a good grasp of the current literature on their topic area (i.e., recent literature and seminal texts relevant to their

study is not cited/referenced).

Figures 1-27 are important to explore the specifics of the paper. Some conclusions contribute to the study of the problem.

Authors need to add more details on the range of simulation considered in this work should be clearly outlined within the abstract. The current statements are vague and too general to get an idea of the work that have been accomplished.

Authors need to add more details on this particular works within citations [25-27].

Publication of this paper seems likely in Materials.

Author Response

September 23, 2020

Dear Reviewer,

Thank you for your comments concerning our manuscript entitled “Numerical simulation and experimental study on residual stress in curved surface forming of 12CrNi2 alloy steel by laser melting deposition” (Manuscript ID: materials-934554). All of those comments are quite significant to our current research and also valuable and helpful for revising and improving our paper. We have studied the comments seriously and made correction which we hope meet with approval. The revisions have be clearly highlighted using the "Track Changes" function in Microsoft Word in the paper. The main corrections in the paper and the responses to reviewer’s comments are as following:

Response to the Reviewer’s comments:

Comments are detailed as follow:

The work is original and relevant to scope of journal because it considered the comparative analysis of numerical simulation and experimental study on residual stress in curved surface forming of 12CrNi2 alloy steel by laser melting deposition. It is conducted with the help of new empirical methods - Finite Element Model. The research method are explained fully. The results are properly discussed and compared to the previous studies. Data is included and properly analyzed. The paper possesses a proper form of well-structured and readable technical language of the field and represents the expected knowledge of the journal`s readership. Some Suggestions are as follows:

  1. 1. In this paper, authors used 39 sources, containing both historical and fundamental works, as well as the latest scientific research on this topic. But the literature review can be improved.

Response: Thank you for your comments. We have carefully read the references (i.e., Global Indicators of Sustainable Development: Evaluation of the Influence of the Human Development Index on Consumption and Quality of Energy) you recommended and have gained a lot. This literature has been added to the literature review, and the specific modified content is shown in the revised document.

Before revision:

It is the preparation material for the camshaft of nuclear power emergency diesel engine [5-6]. Camshaft as an important component of nuclear power emergency diesel generator, its performance and service life play a vital role in nuclear power safety [7-8].

After revision:

The thermal cycles of DLD 12CrNi2 steel were simulated using the finite element method (Abaqus 2017, Dassault, American).It is the preparation material for the camshaft of nuclear power emergency diesel engine [5-6]. As an important energy source, nuclear power plays an important role in human development and sustainable development[7]. Camshaft as an important component of nuclear power emergency diesel generator, its performance and service life play a vital role in nuclear power safety [8-9].

[7]Yumashev, A.; Ślusarczyk, B.; Kondrashev, S.; Mikhaylov, A. Global indicators of sustainable development: evaluation of the influence of the human development index on consumption and quality of energy. Energies, 2020, 13.

  1. 2. The current study brings many new to the existing literature or field. For one, the author(s) seem to have a good grasp of the current literature on their topic area (i.e., recent literature and seminal texts relevant to theirstudy is not cited/referenced).

Response: Thank you for your comments. We have read some recent literature related to this study and added them to the literature review. The detailed changes are as follows:

Before revision:(pages2)

Ding et al. [8] established a multi-layer multi-track finite element model by using ABAQUS software, and studied the effect of different substrate preheating temperature on the residual stress of 12CrNi2 multi-layer multi-track specimen. 

After revision:(pages2-3)

Ding et al. [9] established a multi-layer multi-track finite element model by using ABAQUS software, and studied the effect of different substrate preheating temperature on the residual stress of 12CrNi2 multi-layer multi-track specimen. Kang et al. [26] established a single-layer, double-track finite element model of 24CrNiMo to simulate the temperature field and stress field distribution in the forming process of LMD. Kiran et al. [27] established a single-track, multi-track finite element model of 316L to simulate the temperature field and stress field distribution.

[26] Xueliang, Kang; Shiyun, Dong; Hongbin, Wang; Shixing, Yan; Xiaoting, Liu; Binshi, Xu. Effect of laser power on gradient microstructure of low-alloy steel built by laser melting deposition. Mater. Lett. 2020, 262, 127073.

[27] Kiran, A.; Hodek, J.; Vavřík, J.; Urbánek, M.; Džugan, J. Numerical simulation development and computational optimization for directed energy deposition additive manufacturing process. Materials, 2020, 13, 2666.

  1. 3. Authors need to add more details on this particular works within citations [25-27].

Response: Thank you for your comments. The detailed changes are as follows:

Before revision:(pages2-3)

The above studies are mainly aimed at titanium alloy[25-27], nickel-based alloy[28-29], stainless steel and other materials[30], but there are relatively few studies on 12CrNi2 high-performance alloy steel, which plays an important role in national economy and national defense.

After revision:(pages2-3)

Gharbi et al. [28] studied the DMD process using an Yb-YAG disk laser and a widely used titanium alloy (Ti–6Al–4V) to understand the influence of the main process parameters on the surface finish quality. Qu et al. [29] fabricated Ti-47Al-2.5V-1Cr intermetallic alloy by the laser melting deposition (LMD) manufacturing process, and studied the microstructure by OM, SEM, TEM and XRD. Cottam et al. [30] investigated the effect of deposition rate on the residual stresses formed during the laser cladding of Ti-6Al-4V  powder onto a Ti-6Al-4V substrate. The above studies are mainly aimed at titanium alloy[31], nickel-based alloy[32-33], stainless steel and other materials[34], but there are relatively few studies on 12CrNi2 high-performance alloy steel, which plays an important role in national economy and national defense.

  • Miao, X.; Wu, M.; Han, J.; Li, H.; Ye, X. Effect of laser rescanning on the characteristics and residual stress of selective laser melted titanium Ti6Al4V alloy. Materials, 2020, 13, 3940.

Thank you for your comments.

We have tried our best to improve the manuscript and made some changes in the manuscript, concluding the grammar improvement and some supplementary on content. We appreciate for Reviewer’s warm work earnestly, and hope that the correction will meet with approval. Please feel free to contact us with any questions and we are looking forward to your consideration.

Once again, thank you very much for your comments and suggestions.

Best regards.

Yours sincerely,

Shiyun. Dong

National Key Laboratory for Remanufacturing, Army Academy of Armored Forces, Beijing, 100072, China  

21 Dujiakan, Fengtai District, Beijing

Mobile: (0086)13911655167

E-mail: syd422@sohu.com

Reviewer 2 Report

Although there are works aimed in other alloys, there are relatively few studies on the presented alloy, used is strategic applications, being deserved to be to analysed.

Although there are experimental and numerical part, more comparison points/variables could be interested to include in the paper.

Some particular points are:

  • Has the authors analyse the mesh size effect of the deposition layer?
    • Is not considered element size variation through the thickness?
    • If deposition element thickness is 0.3mm, shouldn’t it be of same thickness at the substrate surface?
  • Pag5: Which is the origin of the heat-flux laws (eqs1&2)?
    • After parameters definition, they collapse in 1 expression. Why this simplification?
    • The change in geometry is agree with LDM movement?
  • PAG7: The Stephen-Boltzmann coefficient is represented by sigma, used also for stress. Although they are properly defined, I suggest to use different “letters” for different “meanings” in papers.
  • It is assumed that all process is in quasiestatic conditions and are not inertial effects, therefore strain rate are not included. Is that correct?
  • Table 2.why is included “error” in third column?
  • Are the thermal parameters compared with literature values?
  • Section 4.1.2.
    • Which tensile machine/characteristics were used?
    • Which was the displacement velocity?
    • I do not have experience with the mentioned extensometer: Is it able to support these high temperatures?
    • I suggest to include stress-strain curve at room temperature for reference (as you do in Fig10b).
    • How many specimens per temperature were tested?  Are mean values/graphs or particular results?
  • The paragraph is correct but maybe could be excluded from text because is basic (authors’ decision). If it remains in document, please express Fig. in English.
  • Last paragraph. Typo after figure 10.
  • Fig11 shows failure section closing to one specimen extreme as temperature increases instead close to mid-span.
    • This happen in all cases? could this affect to the extensometer accuracy?
    • Which is the explanation?
  • Pag15: can be simplifyed the numerical model aplying some kind of symmetry?
  • Fig20: has been analysed these effects at different thicknesses?
  • One of the main remarked point is the curved geometry, therefore:
    • can compare any result, i.e. residual stress, with an equivalent plane specimen?
    • Can be extracted some conclusion as function of the diameter?
    • Can be compared some residual stress with experimental result?

Author Response

September 23, 2020

Dear Reviewer,

Thank you for the reviewer’s comments concerning our manuscript entitled “Numerical simulation and experimental study on residual stress in curved surface forming of 12CrNi2 alloy steel by laser melting deposition” (Manuscript ID: materials-934554). All of those comments are quite significant to our current research and also valuable and helpful for revising and improving our paper. We have studied the comments seriously and made correction which we hope meet with approval. The revisions have be clearly highlighted using the "Track Changes" function in Microsoft Word in the paper. The main corrections in the paper and the responses to the reviewer’s comments are as following:

Response to the Reviewer’s comments:

Comments are detailed as follow:

The paper is well organized and the figures are readable and of good resolution. Although there are works aimed in other alloys, there are relatively few studies on the presented alloy, used is strategic applications, being deserved to be to analysed.

Although there are experimental and numerical part, more comparison points/variables could be interested to include in the paper.

There are some questions the reviewer would like to discuss with the authors.

  1. 1. Has the authors analyse the mesh size effect of the deposition layer?

Is not considered element size variation through the thickness?

If deposition element thickness is 0.3mm, shouldn’t it be of same thickness at the substrate surface?

Response: Thank you for your comments. In finite element analysis, more accurate simulation results can usually be obtained with a finer mesh. However, a finer grid means more elements at the same model size, which consumes more computer resources and computing time. In order to improve the calculation accuracy and reduce the calculation time, this paper adopts the non-uniform mesh for the whole model. The small elements size are adopted in the deposited layer area with drastic temperature change, and the element size gradually increases in the area far from the deposited layer. The reason for using this method is that the laser energy is concentrated, the heat affected area is small, and the mesh density of the deposited area can meet the calculation requirements. Meanwhile, the mesh density of the area outside the deposited layer is small, so that accurate results can be obtained at a lower calculation cost. The smaller mesh size provides similar results for the test simulation run, indicating that the selection of the current mesh size is sufficient. At the same time, the thickness of the mesh on the substrate surface is smaller than that of the deposited layer, which is to obtain the size profile of the molten pool after simulation more accurately and ensure the simulation accuracy. The mesh thickness of the substrate surface is not required to be consistent with the deposited layer.

Related similar applications have been reported in many literatures, such as references [9,21].

[9] Ding, C.; Cui, X.; Jiao, J.; Zhu, P. Effects of substrate preheating temperatures on the microstructure, properties, and residual stress of 12CrNi2 prepared by laser cladding deposition technique. Materials 2018, 11(12).

  • Nazemi,N.; Urbanic, J.; Alam, Hardness and residual stress modeling of powder injection laser cladding of P420 coating on AISI 1018 substrate. Int. J. Adv. Manuf. Tech. 2017, 93(9-12):3485-3503.
  1. 2. Pag5: Which is the origin of the heat-flux laws (eqs1&2)?

After parameters definition, they collapse in 1 expression. Why this simplification?

Response: Thank you for your comments. Laser melting deposition is a relatively difficult numerical simulation process. In the simulation of temperature field and stress field, it is very important to set up and load the heat source function. Generally, the heat source is divided into body heat source and surface heat source. However, among many heat source models, although gaussian surface heat source is widely used as a model cited in many literatures, it cannot meet the simulation requirements for multi-track multi-layer complex shape deposition due to the shallow depth of melt pool. Therefore, the body heat source model has been more and more applied and paid attention to.

Typical functional body heat sources are gauss rotating body heat sources (also regarded as vertebral body heat sources) and double ellipsoid heat sources, as well as some combined heat sources (such as Gauss surface heat sources combined with cylinder heat sources). These volumetric heat sources have been verified in some authoritative literatures, and their simulation effects are indeed good at ordinary Gaussian surface heat sources. In this paper, the double ellipsoid heat source is selected as the initial heat source, and make changes on this basis. In this paper, the heat flow law (formula 1 and Formula 2) originated from the hemispheric body heat source model, then evolved into the ellipsoid body heat source model, and finally formed the double ellipsoid body heat source model. The double ellipsoid heat source model has four shape parameters, which can accurately fit the size of molten pool. However, due to many parameters, it is more difficult to check the parameters. In this paper, considering that the laser spot is circular, and in order to simplify the verification process of the surface heat source model, the formula is combined and transformed into an ellipsoidal heat source model, which can not only ensure the calculation accuracy, but also ensure that the heat source in rectangular coordinate system can be successfully transformed to the heat source in cylindrical coordinate system, thus improving the calculation efficiency.

  1. 3. The change in geometry is agree with LDM movement?

Response: Thank you for your comments. The change of geometric shape is consistent with the motion of LMD. The scan track described in Figure 1 (b) in the paper.

  1. 4. PAG7: The Stephen-Boltzmann coefficient is represented by sigma, used also for stress. Although they are properly defined, I suggest to use different “letters” for different “meanings” in papers.

Response: Thank you for your comments. The symbol in this paper is the content described in the reference [a], but I quite agree with your suggestion. Finally, the symbol is changed to  and the specific content has been modified in this paper.

[a] Gu, D.; He, B. Finite element simulation and experimental investigation of residual stresses in selective laser melted Ti–Ni shape memory alloy. Computational Materials Science, 2016, 117:221-232.

  1. 5. It is assumed that all process is in quasiestatic conditions and are not inertial effects, therefore strain rate are not included. Is that correct?

Response: Thank you for your comments. This opinion has made me think a lot. I may not have given an accurate answer. I hope you can understand. The quasi-static state can also be regarded as the equilibrium state. For example, the velocity of the force applied in the tensile test is extremely slow, so the specimen can be said to work under the quasi-static state, and the influence of strain rate can be ignored. At the same time, according to the previous research results of our research group (ref. 44), under the technological parameters of this experiment, the sample prepared has a high density and no internal pores, inclusions and other defects.

[44] Xuan, Zhao; Yaohui, Lv; Shiyun, Dong; Shixing, Yan; Xiaoting, Liu; Yuxin, Liu; Peng, He; Tiesong, Lin; Binshi, Xu; Hongsheng, Han. The martensitic strengthening of 12CrNi2 low-alloy steel using a novel scanning strategy during direct laser deposition. Opt. Laser Technol. 2020, 132:106487.

  1. 6. Table 2.why is included “error” in third column?

Response: Thank you for your comments. The "error" in Table 2 is a spelling error and has been removed from the text.

  1. 7. Are the thermal parameters compared with literature values?

Response: Thank you for your comments. In this paper, the thermal parameters are tested by preparing samples, and the test results are convincing.

  1. 8. Section 4.1.2.

Which tensile machine/characteristics were used?

Which was the displacement velocity?

I do not have experience withthe mentioned extensometer: Is it able to support these high temperatures?

Response: Thank you for your comments. In order to test the high temperature mechanical properties of 12CrNi2 alloy steel by laser melting deposition, the high temperature tensile specimens were prepared in conjunction with the testing Center of University of Science and Technology Beijing. The high temperature tensile tests were carried out according to GB/T 228.2-2015 <<tensile Test of Metallic Materials-part 2: high temperature test method>>. This is half a year ago, it is difficult to recall the details, only remember the required parallel length of 40mm, the length of the extension gauge distance of 25mm, the test temperature to 900℃. Thank you very much for your comments. I will have a further understanding later.

  1. 9. I suggest to include stress-strain curve at room temperature for reference (as you do in Fig10b).

Response: Thank you for your comments. However, due to the epidemic situation, it is difficult to complete the experiment at room temperature in a short period of time, which will be further analyzed in the later stage.

  1. 10. How many specimens per temperature were tested?  Are mean values/graphs or particular results?

Response: Thank you for your comments. Three samples were tested at each temperature and the stress-strain curve was plotted by taking the mean value.

  1. 11. The paragraph is correct but maybe could be excluded from text because is basic (authors’ decision). If it remains in document, please express Fig. in English.

Response: Thank you for your comments. The picture has been modified, as shown in Figure 9.

Before revision:

Figure 9. Real stress-strain curve.

After revision:(pages2-3)

Figure 9. Real stress-strain curve.

  1. 12. Last paragraph. Typo after figure 10.

Response: Thank you for your comments. The error has been modified in the paper.

  1. 13. Fig11 shows failure section closing to one specimen extreme as temperature increases instead close to mid-span.This happen in all cases? could this affect to the extensometer accuracy?Which is the explanation?

Response: Thank you for your comments. In this experiment, most of the tensile samples were broken in the cylindrical region in the middle of the tensile samples. For all the samples, the fracture position was closer to the clamping end with the increase of temperature.This is because the tensile force on the cylinder in the middle of the tensile specimen is equal, and the fracture position is random.

  1. 14. Pag15: can be simplifyed the numerical model aplying some kind of symmetry?

Response: Thank you for your comments. If only for the temperature field, the symmetric model can be used to simplify it.

  1. 15. Fig20: has been analysed these effects at different thicknesses?

Response: Thank you for your comments. According to the analysis, with the increase of the print layer thickness, the stress of each layer tends to increase first and then decrease, which is because the temperature gradient is large at the beginning of deposition. With the increase of the stress, when the deposition reaches a certain number of layers, the temperature distribution is uniform, the temperature gradient decreases and the stress decreases.

  1. 16. One of the main remarked point is the curved geometry, therefore:

can compare any result, i.e. residual stress, with an equivalent plane specimen?

Response: Thank you for your comments. Curved surface stress is mainly represented by circumferential stress, radial stress and axial stress. Plane stress is mainly divided into transverse stress and longitudinal stress, which are two different expressions and applied in different working environments.

  1. 17. Can be extracted some conclusion as function of the diameter?

Response: Thank you for your comments. I may not understand the question very well.In the direction of diameter, I have also analyzed some data, but the obtained value is very small, far less than the yield strength of the material, and there is no obvious change rule. Compared with the influence of axial stress and circumferential stress, it is very small.

  1. 18. Can be compared some residual stress with experimental result?

Response: Thank you for your comments. In the preliminary experimental design stage, the residual stress test was taken into account, but due to the epidemic, the experiment was difficult to carry out. In this paper, the material parameters have been tested to ensure the reliability of the simulation parameters. Meanwhile, the heat source has been checked by combining simulation and experiment, and corresponding support has been provided for the simulation results. The laws found in the paper are convincing.

Thank you for your comments.

We have tried our best to improve the manuscript and made some changes in the manuscript, concluding the grammar improvement and some supplementary on content. We appreciate for Reviewer’s warm work earnestly, and hope that the correction will meet with approval. Please feel free to contact us with any questions and we are looking forward to your consideration.

Once again, thank you very much for your comments and suggestions.

Best regards.

Yours sincerely,

Shiyun. Dong

National Key Laboratory for Remanufacturing, Army Academy of Armored Forces, Beijing, 100072, China  

21 Dujiakan, Fengtai District, Beijing

Mobile: (0086)13911655167

E-mail: syd422@sohu.com
